# Reaching to inhibit a prepotent response: A wearable 3-axis accelerometer kinematic analysis

**Alessia Angeli**[1‡], **Irene Valori**[2‡], **Teresa Farroni**[2]*, **Gustavo Marfia**[3]

**1** Department of Computer Science and Engineering, University of Bologna, Bologna, Italy, **2** Department of Developmental Psychology and Socialisation, University of Padova, Padova, Italy, **3** Department for Life Quality Studies, University of Bologna, Rimini, Italy

‡ These authors contributed equally to this work and share first authorship.
* teresa.farroni@unipd.it

**Data Availability Statement:** All data files are available from the OSF public repository at the following URL: https://osf.io/37krn/?viewonly= a82c6829d6964d4a8c593ae524ac9e12598.

## Abstract

The present work explores the distinctive contribution of motor planning and control to human reaching movements. In particular, the movements were triggered by the selection of a prepotent response (Dominant) or, instead, by the inhibition of the prepotent response, which required the selection of an alternative one (Non-dominant). To this end, we adapted a *Go/No-Go* task to investigate both the dominant and non-dominant movements of a cohort of 19 adults, utilizing kinematic measures to discriminate between the planning and control components of the two actions. In this experiment, a low-cost, easy to use, 3-axis wrist-worn accelerometer was put to good use to obtain raw acceleration data and to compute and break down its velocity components. The values obtained with this task indicate that with the inhibition of a prepotent response, the selection and execution of the alternative one yields both a longer reaction time and movement duration. Moreover, the peak velocity occurred later in time in the non-dominant response with respect to the dominant response, revealing that participants tended to indulge more in motor planning than in adjusting their movement along the way. Finally, comparing such results to the findings obtained by other means in the literature, we discuss the feasibility of an accelerometer-based analysis to disentangle distinctive cognitive mechanisms of human movements.

## Introduction

Our everyday life is deeply defined by the voluntary actions we execute towards ourselves and towards the world that surrounds us. The way we plan and control our movements has been widely investigated for different motor tasks, to deepen our understanding of which motor strategies individuals adopt to select and execute different goal-oriented actions. In particular, as action features are usually movement-specific, this work focuses on a specific arm movement, namely reaching, which allows human beings to act within their peri-personal space by grasping, manipulating and using objects, as well as to interact with their own bodies and with other people.

**Funding:** T.F. was funded by the University of Padua (https://www.unipd.it/en/), with a SID grant, and by the Beneficentia Stiftung Foundation (http://www.beneficentia-stiftung.ws/en/); G.M. was funded by the University of Bologna (https://www.unibo.it/en/homepage), with the Alma Attrezzature 2017 grant, and by the Golinelli Foundation (https://www.fondazionegolinelli.it/en), with the Data Science scholarship. The funders had no role in study design, data collection and analysis, decision to publish, or preparation of the manuscript.

**Competing interests:** The authors have declared that no competing interests exist.

Performing this specific action requires both a pre-planning and an on-line control of the desired motor output. Such two mechanisms are settled in distinct brain regions, respectively intervene in either the early or later movement time and appear influenced by different sensorimotor aspects and cognitive processes [1]. Indeed, the role of motor networks might go beyond the action specification that answers to the "how to do it" and contribute to the simultaneous process of action selection, which addresses the "what to do" issue and chooses among currently available options [2]. It goes without saying that cognitive control is fundamental to the process of action selection, including the ability to inhibit inappropriate or incorrect responses [3]. Rather than an unitary process, inhibition is a multifaceted skill that comprehends sensory, cognitive, behavioural and motor sub-components [4], such as the ability to stop prepotent motor activities.

In neuro-psychology, one of the most commonly used task to assess motor inhibition of prepotent repsonses is the *Go/No-Go* paradigm [5]. The "Go" trials require participants to provide a fast response (i.e., do something) as soon as a dominant cue appears. On the other hand, the "No-Go" trials require to inhibit the response and not answer (i.e., do nothing) when another non-dominant cue appears (the latter usually appears less frequently than the dominant one) [6]. However, the classical task is unable to investigate the different motor strategies individuals may adopt to perform either a prepotent or an alternative response. To further distinguish between planning and control aspects, kinematic measures have been included with adapted *Go/No-Go* paradigms that asked participants to perform either a prepotent action elicited most of the time (dominant), or an alternative less frequent one (non-dominant). In one adaptation of the *Go/No-Go*, both Reaction Time (RT) and Movement Duration (MD) were analysed, whereby the non-dominant action might be performed with a longer RT or a longer MD depending on whether the actor required either a longer planning phase before the movement onset or a greater control and adjustment during its execution [7].

Nevertheless, it is worth noting that motor planning is not relegate to RT but also overlaps with motor control during the MD. Indeed, "as planning is generally operative early and control late in a movement, the influence of each will rise and fall as the movement unfolds" [1, p. 5]. Therefore, kinematic indices other than RT and MD would be more informative to further clarify the mechanisms beneath distinct movements, with promising possibilities to distinguish the specific inhibitory impairments that are common of several neuro-psychological conditions [8]. As planning seems to be primarily devoted to process cognitive information, whereas control is dedicated to homing in on a target with specific spatial features [1], the inhibition of prepotent motor responses evoked by *Go/No-Go* tasks would likely load on planning mechanisms.

The movement research field has extensively debated regarding the distinctive meaning of different motor indices, which are affected by different factors, thus providing insights on distinct neuro-psychological mechanisms underlying motor activities. Acceleration, in particular, discloses the movement smoothness, whereby an optimal reaching is ideally (for instance in experimental contexts and robotics) the one with the minimum jerk, namely, the rate of acceleration change in time [9, 10]. The smoothness of a reach-to-grasp movement might depend on whether the target object is present, imagined or absent, on how it is oriented, or on which is the plane of movement (e.g., horizontal or vertical plane) [11].

Neuro-imaging studies collected evidence of distinct cortical networks being related to distinct kinematic features. Bourguignon et al. [12] studied the fast repetitive voluntary hand movements of neuro-typical adults and revealed that movement acceleration was mainly coupled with a coherent activation of contralateral primary motor (M1) hand area at $\approx$ 3 Hz and $\approx$ 6 Hz of movement frequencies. Moreover, only when the hand movement aimed at touching its own fingers, the primary somato-sensory (S1) hand area became the most coherent

brain area at ≈ 3 Hz of motion frequency. In addition, the activation of DLPFC (dorsolateral prefrontal cortex, which is responsible for goal-directed action planning) and PPC (posterior parietal cortex, which is responsible for sensorimotor integration and movement monitoring) areas were coherent with movement acceleration [13].

Focusing on velocity, the minimum-jerk model predicts that reaching trajectories starting and ending at full rest will show a symmetric, bell-shaped velocity path, with 50% of MD spent both accelerating and decelerating. However, MD and velocity across time are shaped by several factors, such as the individual developmental trajectory [14], the affordances of the target object (e.g., a cup or a spoon) [11], and social intentions during interactions with others [15]. On this matter, the Time to Peak Velocity percentage (TPV%) is a relative asymmetry index whereby the ideal symmetrical value of 50% would indicate an equivalent acceleration/deceleration phase. Given that whether a kinematic parameter occurs earlier or later over the MD would reflect more either planning or control [1], a small TPV% resulting in a longer deceleration phase may indicate a greater need for control and adjustment of the ongoing movement. On the other hand, a big TPV% resulting in a shorter deceleration phase may indicate a greater need for motor planning.

### Aims and hypotheses

The present work aims at disentangling the contribution of motor planning and control in the selection or inhibition of a prepotent reaching movement. To effectively inhibit a prepotent response and select an alternative one, neuro-typical adults might employ different strategies, such as devoting more time to either the first (planning) or last (online control) movement phases. Therefore, we aim at studying the timing of the peak velocity (that usually occurs at around 40% of duration in voluntary reaching movements [16]) in participants' performance at the *dominant* and *non-dominant* conditions of an adapted *Go/No-Go*. From an exploratory perspective, we expect accurate motor inhibition to result in either a bigger TPV% when primarily built on motor planning, or a smaller TPV% when mainly derived from the control and adjustment of the ongoing movement [1]. Moreover, we aim at using a low-cost portable motion tracking tool, to boost the applicability of our methods to a broad range of research and clinical contexts.

## Materials and methods

### Participants

For this study, we recruited 19 neuro-typical adults aged from 18 to 26 years old (M = 22.3, SD = 1.9), among them 5 men. Recruitment took place among university students with no past or present history of clinical conditions (self-reported). They voluntarily participated in the study and did not receive compensation.

### Procedure

Participants were welcomed into the lab and asked to sign a written consent form. The study was approved by the Ethics Committee of Psychology Research, University of Padua.

Participants sat on a desk and wore an accelerometer research watch on their dominant wrist (the experimental set-up is depicted in Fig 1). They were then asked to place the dominant hand at a specific starting position, monitored by a presence sensor. At the distance of their arm length, they found a response touchscreen so that they were required to completely extend their arm to touch the response screen. A specific task was proposed and required the participant to make action selection choices by touching one of the response keys on the

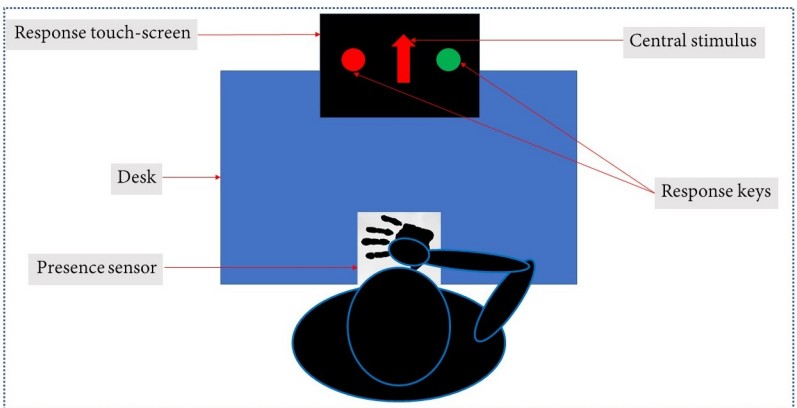

**Fig 1. Experimental set-up.**

screen. The task tested the participant's ability to select a prepotent or an alternative response. During this behavioural task, the kinematics of participant's dominant arm was monitored by the wrist worn 3-axis accelerometer. The task lasted about 15 minutes.

## Apparatus

Although motor analysis is highly informative both in research and clinical settings, kinematic studies often rely on expensive, bulky and sophisticated motion capture systems which may not be affordable in most operative and experimental contexts. In order to use low-cost portable solutions and boost the applicability of motion analysis, both custom made [17] and commercial tools have been recently evaluated. One extensively used commercial option is the Leap Motion Controller system, a small compact device containing two cameras and three infrared light diodes which has, however, spatial and temporal limits compared to motion capture systems [18]. Another commercial possibility that seems more promising in terms of measurement reliability and validity are the inertial sensors built with 3-axis accelerometers, gyroscopes, and magnetometers. In particular, Cahill-Rowley and Rose [19] analysed human reaching kinematics through both inertial sensors and gold standard motion capture systems. The two methods provided consistent measures of displacement, peak velocity magnitude and timing. In light of this encouraging evidence, the time is ripe for the use of low-cost accelerometers to investigate distinct neuropsychological mechanisms beneath action selection.

In the present study, we employed the GENEActiv Original 3-axis wrist worn accelerometer [20] (size: 43 mm × 40 mm × 13 mm, weight without the strap: 16 g) to monitor participants' arm movements. The device measured accelerations through a MEMS sensor, within a range of +/− 8 g, at a 12 bit (3.9 mg) resolution with a 100 Hz logging frequency.

The task was implemented resorting to a JavaFX based application [21].

To run the experiment, we employed a laptop Lenovo G50–80 (Intel Core i5–5200U (2.2 GHz), 4 GB DDR3L SDRAM, 500 GB HDD, 15.6" HD LED (1366 × 768), Intel HD Graphics 5500, Windows 10 64-bit).

The analysis of the resulting data was performed resorting to Python [22] and primarily to "pandas", "numpy" and "scipy" libraries.

Participants responded by tapping on a 19 inch touchscreen (LG-T1910BP, response time 5 ms). The presence-absence of the participant's hand on the starting position was detected through a custom-made presence sensor based on Arduino Leonardo which sent the hand detection data to the laptop via one of its USB ports. It was connected to a ground capacitor

(100 pF) and a capacitive sensor, which consisted of a copper foil wrapped with plastic film (dimension 20 cm × 12 cm, thickness 0.1 mm). The presence sensor program was written using the Arduino Capacitive Sensing Library.

## Task: Inhibition of a prepotent response

A *Go/No-Go* paradigm was adapted to assess the inhibition of a prepotent response. In particular, on the arrival of a central stimulus (red/green, upwards/downwards arrow), participants were asked to select, reach and press one of two response keys (either a red circle or a green circle) placed one on the left and one on the right side of the central stimulus. Participants were told to select the response key of the same color of the central stimulus when it was an upwards/downwards (counterbalanced between participants) arrow (*dominant condition*). On the other side, they were told to select the response key of the different color when the central stimulus was an averted (either upwards or downwards, counterbalanced between participants) arrow (*non-dominant condition*). We built a prepotent response for the same-colour action, given that it was the one that appeared with a higher chance (75%). On the contrary, we elicited an inhibitory different-colour action, which was the less probable one (25%). In this way, we were able to measure the kinematics of dominant vs non-dominant selections, being the movements equal.

Participants were instructed to reply as quickly as possible and had to press any keys within 2,000 ms not to make a response "omission". When participants moved their dominant hand from the starting position before the appearance of the cue stimulus, the response was tagged as "anticipation" and the program aborted the trial by providing no cue stimulus. For this task, each participant was required to perform 160 valid trials (i.e., trial with correct/incorrect answer). In any case, the total trials never exceed a maximum number of 180. Trials were divided into two blocks, distinguished by the red/green response keys being located once on the right and once on the left side of the touchscreen. To maintain participants' engagement during the task, a short (30 seconds on average) video from well-known movies appeared every 40 trials.

Before the start of the next trial, the participant had to return his hand on the sensor. As soon as the hand was in place, as long as the previous trial was not running anymore, the next trial started after a random delay in the range from 0 to 2,000 ms. We will refer to this independent variable as StimulusRandomTime and analyse its effect on participants' performance. Indeed, this variable manipulated the time available to pre-activate the sensorimotor system and predict the incoming occurrence of the central stimulus, potentially affecting the response timing [23].

## Kinematic measures

For each valid trial (i.e., no anticipation, no omission) we reported the following time instants:

$$\text{sensor pressed,} \quad \text{stimulus appeared,} \quad \text{sensor released,} \quad \text{answer given,}$$

which from now on we will refer to as P, S, R, A.

To obtain these data, we synchronized the software logs and the accelerometer with the computer local time, thus combining the accelerometer data with the task outputs.

The time intervals that are related to the kinematic measures of interest were $[S, R]$ which defined RT and $[R, A]$ that corresponded to the MD and was used to compute the TPV%. In addition, the interval $[P, S]$ determined the StimulusRandomTime.

As described in detail in S1 Appendix, the effective acceleration was individuated by means of raw accelerometer data calibration and preprocessing. Subsequently, we computed velocity

and Time to Peak Velocity percentage (TPV%), which is the percentage of time spent from R to maximum peak velocity in the time interval from R to A (i.e., the MD). In the following, we walk through the methodology adopted to compute the TPV% value.

From a theoretical and mathematical point of view, the most direct way to start computing the TPV% is by applying an integration in time and obtain velocity from acceleration. In particular, let $a(t)$ be the acceleration signal on one axis, the related velocity signal $v(t)$ can be computed as follows:

$$v(t) = \int_{t_i}^{t_f} a(dt)dt + C,$$

where $t_i$ and $t_f$ are the initial and final time instants of the movement and $C$ is an integration constant.

However, when facing with real data and numerical functions (e.g., numerical integration), numerical errors can return unreliable velocity values.

Considering the calibrated and preprocessed acceleration (S1 Appendix), let $acc_{RA}$ be the signal related to the time interval $[R, A]$ of a specific valid trial, we applied the cumulative trapezoidal numerical integration function in order to compute velocity. In Fig 2, we reported the velocity components obtained by applying this function to the acceleration values of a trial. After this step, we computed the magnitude (which represents the velocity module) from its components, also shown in Fig 2.

Notably, the application of an integration function could lead to an incremental numerical error due to a possible bias (i.e., additive noise) present in the acceleration, visible in Fig 2, whereby the x component and magnitude of velocity present increasing monotonous curves rather than the expected bell shape. Such phenomenon, may lead to the creation of a "new" and "false" maximum peak at the end of MD, making the computation of the central "true" peak quite challenging. To overcome this issue, we applied the detrend function to

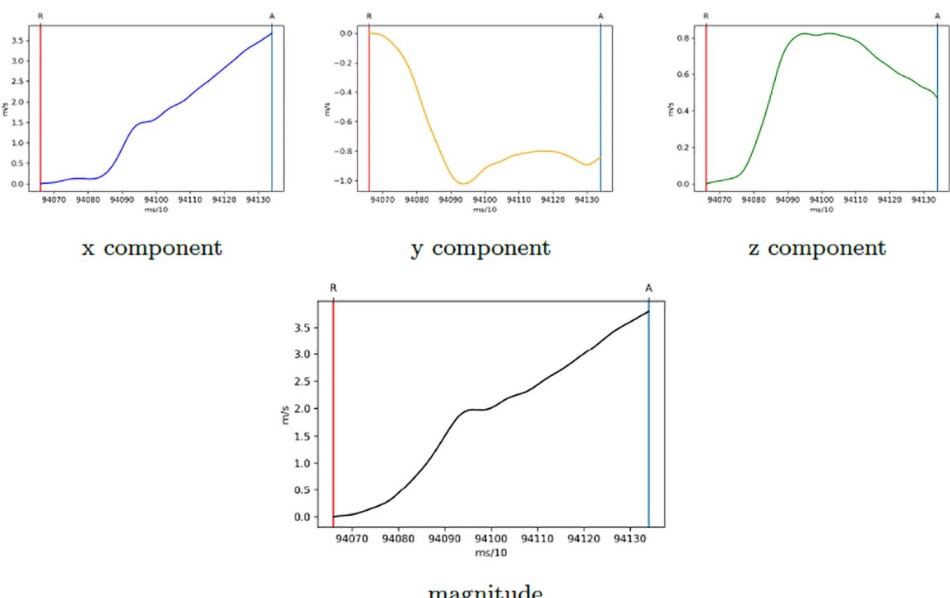

**Fig 2. Velocity signals of a trial where the error due to the acceleration bias is visible in both the x component and the magnitude (increasing monotonous curves that do not represent the expected bell shape).**

the velocity magnitude, thus removing the signal linear trend and reducing the numerical error described above (further details are reported in S2 Appendix). While the velocity values could change due to the detrend function application, the position in time of the peak velocity appeared stable, thus allowing us to calculate the TPV% ("when"). On the other hand, we were not able to further investigate those indices based on the velocity value ("how fast", e.g., mean velocity, value of peak velocity), as supported also by the supplementary analysis described in S3 Appendix.

Ultimately, we aimed to exclude possible extreme TPV% values that would be due to numerical errors, in cases where the detrend function was not sufficient to remove their effect on the signal. Moreover, we aimed to remove those observations with TPV% values that were unlikely related to task-related human reaching movements, but rather potentially ascribable to extra-task movements. For these reasons, the a-priori inclusion criteria for valid TPV% values comprehended those between 5% and 95%. At the end of this procedure, we excluded 59 out of 2, 962 trials.

## Statistical approach

In light of the novelty of our paradigm, an exploratory approach was elected to test different potential hypotheses through a model comparison. We investigated whether the TPV% was influenced by the random effect of participants (i.e., interpersonal variability), as well as the fixed effect of condition (within-subjects, two levels categorical factor: dominant versus non-dominant). Moreover, we checked for the effect of the random time before the central stimulus onset. The latter was a continuous independent variable that we named StimulusRandomTime. Each research hypothesis was specified as a statistical model, such that the statistical evidence of the formalised models was evaluated using information criteria [24].

Mixed-effects models were employed to account for the repeated measures design of the experiment (i.e., trials nested within participants). In particular, generalized mixed-effects models were used considering the Beta distribution (with logit link function) of our dependent variable (TPV%). Indeed, the TPV% contained continuous proportions on the interval (0, 100), easily rescaled in the interval (0, 1) (TPV), and can be approximated by a Beta distribution [25]. The statistical analyses were conducted using the R version 4.0.2 [26], with the "glmmTMB" package [27] to run the model comparison.

To the end of exploring our data, we specified four nested models with the TPV as dependent variable and the random effect of participants:

- `mb0` specified the hypothesis of no difference due to the independent variables and only accounted for the random effect of participants;

- `mb1` specified the hypothesis of a difference due to the condition effect;

- `mb2` specified the hypothesis of a difference due to the additive effect of condition and StimulusRandomTime;

- `mb3` specified the hypothesis of a difference due to the interaction effect of condition and StimulusRandomTime.

The details of the model specification are depicted in Table 1.

Therefore, the four models were compared through the Akaike weights (i.e., the probability of each model, given the data and the set of considered models) [24], using the R package "AICcmodavg" [28]. Moreover, the models were compared using a likelihood ratio test (`anova(mb0, mb1, mb2, mb3)` R function).

**Table 1. Model specification.**

| Model | Dependent variable | Random effect | Fixed effects |
|---|---|---|---|
| mb0 | TPV | Participants | – |
| mb1 | TPV | Participants | Condition |
| mb2 | TPV | Participants | Condition + StimulusRandomTime |
| mb3 | TPV | Participants | Condition × StimulusRandomTime |

**Table 2. Descriptive statistics ($n_{participants}$ = 19).**

| Condition | RT | | | | | MD | | | | | TPV% | | | | |
|---|---|---|---|---|---|---|---|---|---|---|---|---|---|---|---|
| | $n_{trials}$ | min | max | M | SD | $n_{trials}$ | min | max | M | SD | $n_{trials}$ | min | max | M | SD |
| Dominant | 2,253 | 62 ms | 1,373 ms | 558 ms | 136 ms | 2,253 | 266 ms | 1,562 ms | 500 ms | 167 ms | 2,213 | 5.04% | 94.36% | 40% | 15% |
| Non-dominant | 709 | 335 ms | 1,365 ms | 601 ms | 163 ms | 709 | 291 ms | 1,562 ms | 591 ms | 207 ms | 690 | 6.9% | 94.31% | 45% | 17% |

Note: TPV% includes less trials due to the exclusion of extreme values.

# Results

The nineteen participants provided 2,962 correct responses, 54 incorrect ones (24 in the *dominant condition* and 30 in the *non-dominant condition*), 107 omissions (78 in the *dominant condition* and 29 in the *non-dominant condition*) and 22 anticipations. Minimum and maximum values, means and standard deviations of RT, MD and TPV% of correct responses in each condition are reported in Table 2.

The distribution of TPV values in each condition is shown in Fig 3.

The model comparison outputs, namely the degree of freedom (*Df*), the Akaike weights (*AICcWt*), the chi-squared test statistic values ($\chi^2$) and the p-values (*p*) are reported in Table 3.

The most plausible model given the data and the set of considered models was mb2 (*AICcWt* = 0.44), which included the random effect of participants, the additive effects of condition (statistically significant according to *p* < .001) and StimulusRandomTime (statistically non significant according to *p* = .08) (p-values from summary(mb2) R function). These effects, predicted by model mb2, are depicted in Fig 4.

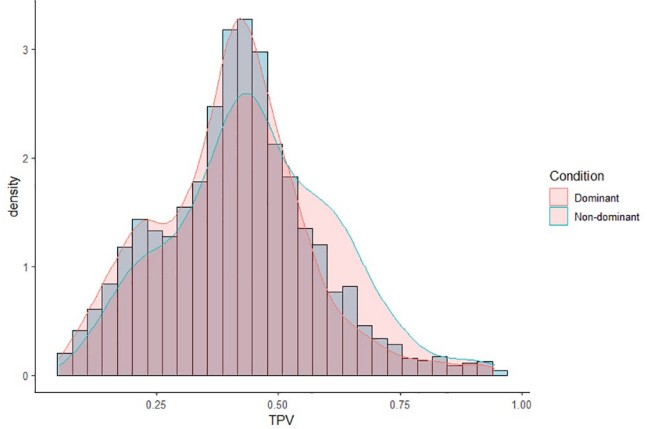

**Fig 3. Distribution of the TPV values ($n_{trials}$ = 2, 903).**

**Table 3. Model comparison.**

| Model | Df | AICcWt | $\chi^2$ | p |
|---|---|---|---|---|
| mb0 | 3 | 0.00 | | |
| mb1 | 4 | 0.20 | 49.70 | <.001 |
| mb2 | 5 | 0.44 | 3.58 | .06 |
| mb3 | 6 | 0.36 | 1.63 | .20 |

## Discussion

The present study investigated the relative contribution of motor planning and control to the inhibition of a prepotent response. We explored neuro-typical adults' movements in a task that required a reaching either to select a prepotent, dominant response or to inhibit the dominant and select the non-dominant alternative. The descriptive statistics indicated that participants performed the non-dominant response (compared to the dominant one) by increasing both the RT (time devoted to motor planning prior to movement onset) and MD (time of motor execution). However, these two indices are not sufficient to disentangle the planning and control phases of the movement. Indeed, given that motor planning and control overlap during the MD [1], we analysed the Time to Peak Velocity (TPV) to further distinguish these two mechanisms. As a relative asymmetry index, whether the TPV occurred earlier or later over the MD would reflect more either planning or control. From our exploratory model comparison, we can expect people to show bigger TPV in the non-dominant compared to the dominant condition. This evidence supported the idea that adults require a greater motor planning rather than online adjustment to inhibit a prepotent response, select and perform an alternative one. Our results are consistent with the extant literature, whereby planning is devoted to process cognitive information and control is dedicated to get on a target and adjust to its specific spatial features [1].

In addition, the most plausible model given our data and set of specified models showed that when people had to wait more to start the trial (StimulusRandomTime), they increased the movement time devoted to motor planning. Although not significant from a statistical point of view, this effect suggests that a longer preparation time before the trial to start might

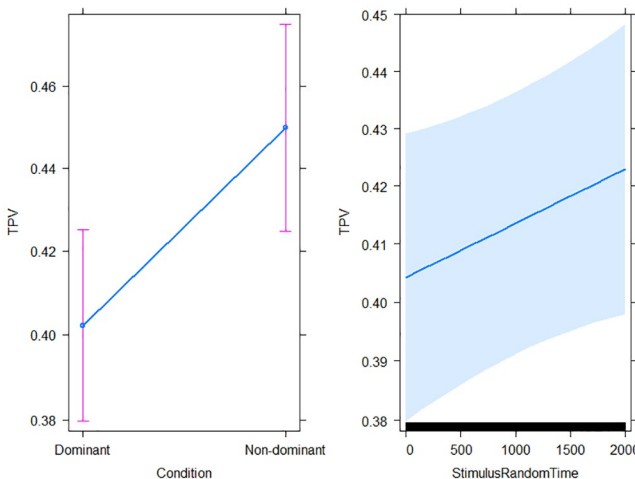

**Fig 4. Model mb2: Condition and StimulusRandomTime effects on the TPV ($n_{participants}$ = 19, $n_{trials}$ = 2, 903).**

allow participants to increase the time devoted to motor planning. We can interpret this finding in light of the massive literature about the preparatory effect of the foreperiod, which is the time from a warning signal and a "Go" stimulus, and is known to affect response times [29]. In our study, participants had to place their hand on the presence sensor to signal their readiness to start the next trial. The time instant they pressed the sensor can be seen as an active warning signal that pre-activate the sensorimotor system. After a variable random time interval, the central "Go" stimulus appeared to trigger participants' response. We can speculate that, within 2,000 ms, a longer preparation time increases adults' motor planning. As the foreperiod effect and the temporal preparation abilities change across development, future studies could expand on the ontogeny of these mechanisms [30].

The present work also employed a low-cost wearable 3-axis accelerometer to investigate human motor inhibition. The inertial sensors built with 3-axis accelerometers, gyroscopes, and magnetometers have been indicated as promising commercial tools to study the kinematics of human movements and overcome the constraints of expensive motion capture systems. Although they have the potential of being portable and wearable, they appeared to provide accurate and reliable data only for some kinematic indices, such as the value and timing of peak velocity [19]. Based on our kinematic measurements and analyses, the kinematic indices built upon the velocity value did not appear sufficiently reliable and valid (as reported in S3 Appendix). On the other hand, those related to the velocity shape over time seemed to be valid indices. Indeed, our average Time to Peak Velocity percentage (TPV%) was consistent with those reported by previous studies, similar tasks and motion capture systems with highest level of precision [16]. Therefore, we support the use of a commercial and low-cost 3-axis accelerometer to calculate the TPV% and compare participants' performance.

It is worth mentioning that the present study has some limitations. Firstly, our sample did not include a balanced number of women and men, thus preventing us to make any claims about potential gender differences that should be furthered in future studies. Secondly, we could not base our sample size specification on previous literature that tested motor inhibition through the TPV%. Therefore, our findings should be interpreted as preliminary and exploratory indications to develop future confirmatory studies. Moreover, future studies might include video recordings and offline coding of the experimental sessions, thus checking for potential cases where participants show extra-task movements that could result in anomalous trials. Ultimately, from a methodological point of view, to further increase the accuracy of the preprocessing, in particular to remove the gravity component from the acquired acceleration, future studies could use a combination of accelerometer and gyroscope. In this way, data related to the orientation of the accelerometer would be available in order to remove the gravitational acceleration. However, the gyroscope would not solve the numerical errors driven by possible accelerometer bias and numerical mathematical functions. These issues could be addressed from an algorithmic point of view, with the evaluation of other methods and models in order to process raw accelerometer data in a way that could reduce the numerical errors. An algorithm class that could obtain promising results with huge amount of raw data is the learning class. Machine and deep learning algorithms could study different input signals and learning information from all the data. In this case, a supervised data set would incrementally improve the results but also an unsupervised approach could be taken into consideration.

Overall, this study expands on our understanding of which motor strategy is successful for neurotypical adults to inhibit prepotent reaching movements. This would lay the foundations for investigating the atypical strategies implemented by individuals and clinical groups with inefficient motor inhibition. Although motor inhibition is affected in a number of neurodevelopmental disorders, the underlying multifaceted mechanisms shape unique phenotypes that require appropriate and specific interventions [31]. For instance, inhibitory skills are linked to

individual traits such as impulsiveness [32], and inhibitory control deficits have been found through Go/No-Go tasks in autism spectrum disorders [33], whereby difficulties in inhibiting prepotent responses seem to be associated with higher-order repetitive behaviours [34]. Moreover, inhibition is part of a broader category of control processes named executive functions, which are distinguished but correlated [35], and play a fundamental role in everyday action selection and execution. Indeed, although difficulties and impairments in the action domain are common to several clinical conditions (i.e., multiple sclerosis, Alzheimer's disease, Parkinson's disease), the underlying sensory, motor and cognitive mechanisms might dramatically differ among patients [36–39]. Future studies could utilise the present method and apparatus to disentangle the planning and control mechanisms of motor actions that involve different neuropsychological abilities, thus providing fundamental insights on the design of motor and psychological interventions.

## Supporting information

**S1 Appendix. Acceleration calibration and preprocessing.**
(PDF)

**S2 Appendix. The detrend function application to velocity.**
(PDF)

**S3 Appendix. Reliability and validity of acceleration and velocity values.**
(PDF)

## Acknowledgments

Our gratitude to the clever master students who collaborated to data collection: Sara Pezzotti, Ilaria Rossi, Irene Strappazzon. Many thanks to Andrea Janna for his fundamental technical support and advice.

## Author Contributions

**Conceptualization:** Irene Valori, Teresa Farroni.

**Data curation:** Alessia Angeli, Irene Valori.

**Formal analysis:** Alessia Angeli, Irene Valori.

**Funding acquisition:** Teresa Farroni, Gustavo Marfia.

**Investigation:** Alessia Angeli, Irene Valori.

**Methodology:** Alessia Angeli, Irene Valori.

**Project administration:** Alessia Angeli, Irene Valori, Teresa Farroni, Gustavo Marfia.

**Resources:** Teresa Farroni, Gustavo Marfia.

**Software:** Alessia Angeli.

**Supervision:** Teresa Farroni, Gustavo Marfia.

**Visualization:** Alessia Angeli, Irene Valori.

**Writing – original draft:** Alessia Angeli, Irene Valori.

**Writing – review & editing:** Alessia Angeli, Irene Valori, Teresa Farroni, Gustavo Marfia.

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
