## [Decision Letter · Decision Letter 0]

16 Apr 2021

PONE-D-20-39385

Reaching to inhibit a prepotent response: A wearable 3-axis accelerometer kinematic analysis

PLOS ONE

Dear Dr. Farroni,

Thank you for submitting your manuscript to PLOS ONE. After careful consideration, we feel that it has merit but does not fully meet PLOS ONE’s publication criteria as it currently stands. Therefore, we invite you to submit a revised version of the manuscript that addresses the points raised during the review process.

The manuscript has potential but all concerns raised by both reviewers need to be addressed.

We look forward to receiving your revised manuscript.

Kind regards,

Bernadette Ann Murphy, PhD

Academic Editor

PLOS ONE

Journal Requirements:

2. Please ensure that you have provided sufficient detail on participant recruitment in the Methods section.

*Please also change "female” or "male" to "woman” or "man" as appropriate, when used as a noun (see for instance https://apastyle.apa.org/style-grammar-guidelines/bias-free-language/gender).

3, We note that you have stated that you will provide repository information for your data at acceptance. Should your manuscript be accepted for publication, we will hold it until you provide the relevant accession numbers or DOIs necessary to access your data. If you wish to make changes to your Data Availability statement, please describe these changes in your cover letter and we will update your Data Availability statement to reflect the information you provide.

Reviewers' comments:

Reviewer's Responses to Questions

**Comments to the Author**

1. Is the manuscript technically sound, and do the data support the conclusions?

Reviewer #1: Yes

Reviewer #2: Partly

2. Has the statistical analysis been performed appropriately and rigorously? 

Reviewer #1: Yes

Reviewer #2: I Don't Know

3. Have the authors made all data underlying the findings in their manuscript fully available?

Reviewer #1: Yes

Reviewer #2: Yes

4. Is the manuscript presented in an intelligible fashion and written in standard English?

Reviewer #1: Yes

Reviewer #2: No

5. Review Comments to the Author

Reviewer #1: GENERAL COMMENTS

This a novel study that examined movement kinematics (using a low-cost 3-axis wrist worn accelerometer) within an adapted Go/N-Go task paradigm. Findings revealed that time to peak velocity (TPV%) was higher in the non-dominant (i.e., more challenging trials requiring inhibition) trials when compared to the dominant trials. The authors explain that these findings suggest participants spent more time motor planning rather than engaging in ‘on-line’ control during the movement when performing the non-dominant trials. While this a very simple study design in the sense of what the participants had to actually do, the use of a low-cost 3-axis wrist worn accelerometer to assess movement kinematics (and to assess them relatively accurately) alongside the differences in movement patterns when engaging in an adapted Go-/No-Go paradigm provides fundamental groundwork for future research (at least in my opinion). I can see many ways this research can be built on across various paradigms and populations. However, I found that this was only briefly touched on in the discussion and believe the authors should try to elaborate on points they made for future research and among different populations. For instance, how could this wrist worn accelerometer (and the TPV% findings) be used within skill acquisition settings (e.g., learning in sports or occupational settings) where tasks often require varying degrees of inhibition to perform correctly and/or to avoid performing where a mistake could result in injury. The same applies for additional discussion pertaining to clinical/non-neurotypical populations as well as discussing the findings using ‘simpler terms’ to assist readers who are not as familiar with motor control research (see point #8 below). I also found that the introduction could have been developed a little more so that formal hypotheses could have been stated. There are a couple sentences in the methods (lines 202-207) and results (lines 259-261) that provide indication that the authors could have developed formal hypotheses (which I recommend doing if a revision is submitted). As such, given these concerns and others highlighted below I cannot recommend the paper for publication in its current form.

SPECIFIC CONCERNS

Abstract

1) Sorry if I misunderstood this finding, but I think ‘non-dominant’ should be used in place of ‘dominant’ within this sentence “Moreover, the peak velocity occurred later in time with respect to the dominant response, revealing that participants tended to indulge more in motor planning rather than in adjusting their movement along the way”.

Introduction

2) Line 112 – Stating formal hypotheses around here is recommended based on previous research and in line with what is stated on lines 202-207 and lines 259-261. I also recommend adding an additional paragraph (or 2) prior to this paragraph discussing these points in greater detail.

Materials and methods

3) Participants – Was a sample size calculation performed? If not, how does this number of participants and effect sizes from the present study compare to previous research?

4) Participants – The majority of the sample were female. Should this be discussed as a limitation?

5) Lines 177-178 – I think more information is needed about this short video. For instance, what was contained in this video, how long was it, and how do you know it helps to maintain participants’ attention and engagement?

Results

6) Lines 256-260 and the content contained in Table 1 – Sorry if I missed it, but did the authors conduct any statistical tests comparing these mean values?

Discussion

7) Based on my previous comment, I think the discussion could be improved by including more content pertaining to how future research can build on these findings across different designs and populations.

8) Lines 319-321 – This just an example that could be further developed in other areas of the discussion section, but I was wondering if the authors thought it might be a good idea to break this down even simpler for non-motor control readers that may not be familiar with terms like ‘motor planning’? For example, and I may be incorrect in my interpretation, but stating that people move slower initially when performing a task that requires inhibition suggesting they ‘thought’ longer (or did something cognitively) prior to decelerating or ‘homing’ in on a target. Am I correct to assume there is some sort of additional cognitive (perhaps even unconscious) and/or motor processes that occur during movement tasks requiring some level of inhibition? Similarly, do the findings suggest participants spend longer ‘thinking’ about the correct response after initiating a movement (as they move slower to peak velocity) but then are equally ‘fast’ in the second phase of the movement? Or is it a combination of initiating the movement slower and moving slower once it is initiated?

Reviewer #2: In the present study, the authors examined how two different reaching tasks, a prepotent (dominant) task and an alternate (non-dominant) task, influenced motor planning and online adjustments. Motor planning and online adjustments were inferred from a number of kinematic measures; reaction time (RT), movement duration (MD), and time to peak velocity percentage (TPV%). Parts of these measures were extrapolated from 3-D accelerometer data. The authors found that, while the absolute measures of RT and MD were overall longer during non-dominant reaching, the relative measure of TPV% occurred later in the non-dominant task compared to the dominant. The authors suggest that motor planning was more valuable to participants than online adjustments; the prepotent task had to first be inhibited and an alternate task selected. This is an interesting paper, and I do believe that it adds valuable information to current literature. However, I have some major concerns regarding the current state of the manuscript that I feel should be addressed prior to publication. My major, and minor, concerns are listed below.

Major Concerns

1. My first major concern involves the formatting of the manuscript. In its current form, the manuscript is not presented in traditional manuscript format. The introduction is extremely long, due largely to the fact that it contains a great deal of information better suited to the methods section. The introduction is also lacking hypotheses. There is no mention of statistical analyses in the methods section. Rather, statistical methods are scattered throughout the results section. I normally do not comment on writing style or paper format as this is a largely subjective point that says nothing of the scientific quality of the work. However, in its present form, the manuscript is confusing and difficult to read, making it a challenge to locate pertinent information and decipher the findings of the study. The manuscript needs to be revised to improve readability. Specifically:

a. The introduction should concisely establish a framework related to the current field of work and state the problem/gap that currently exists in the literature. This should be followed by a clear statement of the study’s purpose and the hypotheses. The introduction could be easily cut down to half of its current size.

b. Statistical analyses/tests should be discussed at the end of the methods section, not in the results section. Currently, the majority of the results section are actually method details, not study findings.

2. A second reason why I find the study’s introduction so confusing is that it seems to be trying to accomplish two very different things simultaneously. On one hand, this appears to be a motor planning/motor control study, and on the other, it seems to be a validation study for wearable 3-axis accelerometers. For instance, in the final line of the introduction, the authors state: “In the following, the methods and results, as well as considerations concerning the feasibility, reliability and validity of such an approach are discussed.” However, I don’t see how this is a validation study as the TPV% values were never compared to a gold standard, such as values obtained from motion capture setups. Also, this seems to already have been accomplished by Cahill-Rowley and Rose in 2017 (cited by the study authors), who validated peak velocity magnitude and timing from accelerometers against Motion Analysis Corporation’s setup. Instead, the authors seem to have based the accuracy of their own data on how close it relates to the work of other studies. For example, in the discussion section on lines 344 – 352, the authors make numerous statements such as: “On the other hand, those related to the velocity shape over time seemed to be valid indices.” I am confused by this entire section. It should either be valid or not valid, and if it is valid, what was it validated against? To address this point, the authors should clearly state whether this was a validation study or not. If it is a validation study, how was TPV% validated, and what has this work added to the field that has not already been determined by previous research? (Example, Cahill-Rowley and Rose 2017).

Minor Concerns

3. Although the manuscript is certainly readable in its current form, there are quite a few errors with grammar and sentence structure that should be corrected.

4. Lines 57 – 59: The authors highlight in this sentence that wearable IMUs with built in accelerometers offer promising potential for the future. The Cahill-Rowley and Rose, 2017 paper is referenced here, but no reference is made to what they actually did in their study. The overall introduction seems to be implying that the present study is the first attempt to validate the use of wearable accelerometers, but clearly, papers already exist on this topic. Again, it is not evident to me what the novelty of this study is in regards to the use of wearable accelerometers.

5. Line 63: This is the first time I’ve seen a separate section all to itself included within an introduction, and as stated in my first major comment, I am confused as to why it has been included. Part of this section outlines the concept of investigating the nature of motor planning versus online adjustments using “Go” cue experiments. This could all be included in the main body of the introduction. The rest of the section is spent defining kinematic measures such as RT and MD, which are very standard kinematic measurements that shouldn’t warrant this much text to explain. As stated in my first major comment, the introduction could be easily reduced by 50%.

6. Lines 116-118: As stated in my second major comment, please explain what is meant by validation. The study seems to be trying to validate the use of wearable 3-axis accelerometers, but I don’t see how this validation was accomplished.

7. Lines 121 – 122: An uneven number of females and males participated in this study. Is it possible that sex could affect the kinematic measures, particularly TPV%? Might males and females employ unique strategies when performing a non-dominant task? In other words, is it possible that females might rely more or less on motor planning time than males? The same goes for online adjustments. Was sex accounted for in the statistical tests?

8. Line 170: Please explain why trials longer than 2000 ms were omitted from the study. Given that movement duration was longer for the non-dominant reaching tasks, I assume that more trials were omitted in these tasks than the dominant (prepotent) tasks. If so, would this not diminish the difference between the two reaching tasks?

9. Lines 223 – 225: I may be confused in this, but from what I can gather, the authors initially calculated velocity from acceleration using a trapezoidal integration function. There was a concern that such a function could lead to additional errors if a bias existed. Specifically, it could create a false peak in velocity. Thus, the authors applied a detrend function. As far as I can tell, velocity magnitude was not reported in this study, so why was this detrend function deemed necessary? TPV% was the reported measure, and I don’t believe integration would skew the time domain.

10. Lines 250 – 253: TPV% measures were excluded if they occurred outside of 5 and 95% of the trial’s movement time. The provided rationale is that anything outside of 5 or 95% represents technical failure rather than true human movement. A total of 59 trials were omitted from the present study, indicating that technical failure did in fact occur in some trials. I find this point concerning. How do the authors know that trials within the 5-95% represented true human movement and weren’t also influenced by technical errors/failures? For instance, if I picture myself performing this reaching task myself, I would find values of 6 and 94% equally unlikely to be the true time of peak velocity as values outside of 5-95%.

11. Again, I could be missing this simply because of some odd formatting choices, but were figure captions included anywhere in the initial submission? Typically, these are submitted either directly with the figures themselves, or they are included as a list either before of after the reference list. I didn’t see these included anywhere, which makes interpretation of the graphs difficult. For instance, on lines 223-225, the authors state that the bias is visible in Figure 2, but since no caption is included with the figure, it is not clear what the authors are referring to. The figures do not currently speak for themselves, and I do not see a clear sign of any bias in Figure 2.

6. PLOS authors have the option to publish the peer review history of their article (what does this mean?). If published, this will include your full peer review and any attached files.

Reviewer #1: No

Reviewer #2: No

---

## [Author Response · Author response to Decision Letter 0]

30 Apr 2021

Our sincere thanks to the reviewers and to the editor for this constructive and thoughtful review. We have made a concerted effort to revise the manuscript to address the reviewers’ comments and are glad to return a more coherent and much improved paper.

R: (Reviewer)

A: (Answer)

R1: GENERAL COMMENTS

This a novel study that examined movement kinematics (using a low-cost 3-axis wrist worn accelerometer) within an adapted Go/N-Go task paradigm. Findings revealed that time to peak velocity (TPV%) was higher in the non-dominant (i.e., more challenging trials requiring inhibition) trials when compared to the dominant trials. The authors explain that these findings suggest participants spent more time motor planning rather than engaging in ‘on-line’ control during the movement when performing the non-dominant trials. While this a very simple study design in the sense of what the participants had to actually do, the use of a low-cost 3-axis wrist worn accelerometer to assess movement kinematics (and to assess them relatively accurately) alongside the differences in movement patterns when engaging in an adapted Go-/No-Go paradigm provides fundamental groundwork for future research (at least in my opinion). I can see many ways this research can be built on across various paradigms and populations. However, I found that this was only briefly touched on in the discussion and believe the authors should try to elaborate on points they made for future research and among different populations. For instance, how could this wrist worn accelerometer (and the TPV% findings) be used within skill acquisition settings (e.g., learning in sports or occupational settings) where tasks often require varying degrees of inhibition to perform correctly and/or to avoid performing where a mistake could result in injury. The same applies for additional discussion pertaining to clinical/non-neurotypical populations as well as discussing the findings using ‘simpler terms’ to assist readers who are not as familiar with motor control research (see point #8 below). I also found that the introduction could have been developed a little more so that formal hypotheses could have been stated. There are a couple sentences in the methods (lines 202-207) and results (lines 259-261) that provide indication that the authors could have developed formal hypotheses (which I recommend doing if a revision is submitted). As such, given these concerns and others highlighted below I cannot recommend the paper for publication in its current form.

SPECIFIC CONCERNS

Abstract

1) Sorry if I misunderstood this finding, but I think ‘non-dominant’ should be used in place of ‘dominant’ within this sentence “Moreover, the peak velocity occurred later in time with respect to the dominant response, revealing that participants tended to indulge more in motor planning rather than in adjusting their movement along the way”. 

A: We thank the reviewer for pointing out that this sentence was not completely clear. We believe that specifying “non-dominant” would solve the problem, so that the sentence has now been rephrased as it follows:

“Moreover, the peak velocity occurred later in time in the non-dominant with respect to the dominant response, revealing that participants tended to indulge more in motor planning rather than in adjusting their movement along the way”

R: Introduction

2) Line 112 – Stating formal hypotheses around here is recommended based on previous research and in line with what is stated on lines 202-207 and lines 259-261. I also recommend adding an additional paragraph (or 2) prior to this paragraph discussing these points in greater detail.

A: We greatly appreciate the reviewer’s suggestion and included the paragraph “Aims and hypotheses” that specifies these points. 

R: Materials and methods

3) Participants – Was a sample size calculation performed? If not, how does this number of participants and effect sizes from the present study compare to previous research?

A: We thank the reviewer for touching on this important point. As we used a novel adaptation of the Go/No-Go task, we could not rely on previous studies to anticipate the effect sizes and calculate the sample size accordingly. Therefore, as we state in the statistical approach section (included in the current revision), our study was inherently exploratory. Moreover, to our knowledge there are no other works in literature that tested the TPV% with inhibitory motor tasks. This prevent us from being able to compare our results with the broad literature. Hopefully, future studies will expand on the topic. We now directly address this crucial aspect in the discussion.

R: 4) Participants – The majority of the sample were female. Should this be discussed as a limitation?

A: We appreciate and agree with the reviewer’s concern about potential gender differences, which were not a main interest of the present work. We now acknowledge the unbalanced sample while discussing about limitations. Hopefully, future research will clarify this important aspect.

R: 5) Lines 177-178 – I think more information is needed about this short video. For instance, what was contained in this video, how long was it, and how do you know it helps to maintain participants’ attention and engagement?

A: We thank the reviewer for giving us the chance to better explain the content of the videos. They were 30-seconds (on average) videos of scenes from famous movies, only meant to give the participants engaging breaks. The task description now states:

 “To maintain participants' engagement during the task, a short (30 seconds on average) video from well-known movies appeared every 40 trials.”

We are quite confident that participants paid attention across all trials given the low error rate.

R: Results

6) Lines 256-260 and the content contained in Table 1 – Sorry if I missed it, but did the authors conduct any statistical tests comparing these mean values?

A: Our apologies for the lack of clarity in discussing these data. We clarified that we run statistical tests only on the TPV dependent variable, that was our main interest in light of the theoretical background. We believe that the clear specification of “aims and hypotheses”, and “statistical approach” sections would make this point much clearer.

R: Discussion

7) Based on my previous comment, I think the discussion could be improved by including more content pertaining to how future research can build on these findings across different designs and populations.

A: We completely agree with the reviewer’s suggestion on this point and furthered the discussion pertaining clinical populations. Although we are not into those fields, we appreciate your previous hints on “skill acquisition settings (e.g., learning in sports or occupational settings)” and look forward further applications of the evidence we present in our work.

R: 8) Lines 319-321 – This just an example that could be further developed in other areas of the discussion section, but I was wondering if the authors thought it might be a good idea to break this down even simpler for non-motor control readers that may not be familiar with terms like ‘motor planning’? For example, and I may be incorrect in my interpretation, but stating that people move slower initially when performing a task that requires inhibition suggesting they ‘thought’ longer (or did something cognitively) prior to decelerating or ‘homing’ in on a target. Am I correct to assume there is some sort of additional cognitive (perhaps even unconscious) and/or motor processes that occur during movement tasks requiring some level of inhibition? Similarly, do the findings suggest participants spend longer ‘thinking’ about the correct response after initiating a movement (as they move slower to peak velocity) but then are equally ‘fast’ in the second phase of the movement? Or is it a combination of initiating the movement slower and moving slower once it is initiated?

A: We thank the reviewer for his/her thoughtful reflection. We agree we the need for simplicity to engage also non-motor control readers and we tried to do so. Going into the specifics of your comments, we believe that saying “people move slower” is not appropriate in our case. Indeed, as we highlight in the discussion and further explain in S3 Appendix, we could not rely on velocity magnitudes (“how fast”). 

“the kinematic indices built upon the velocity value did not appear sufficiently reliable and valid (as reported in S3 Appendix). On the other hand, those related to the velocity shape over time seemed to be valid indices.”

Consequently, we could not distinguish the velocity magnitude in the first and second movement phases, but we can only say whether the peak velocity occurred earlier/later within the movement time. This would tell us whether participants spent more time in the first or second movement phase (i.e., at the beginning or at the end of the movement). Moreover, as the reviewer correctly mentioned, our experimental conditions are designed to elicit motor inhibition. In addition to this motor-cognitive mechanism, we cannot say whether people “think more”, which stands for explicit cognitive reasoning. We did not assess participants’ strategies and cannot establish the extent to which they were implicit/explicit and varied across participants. It is also correct to wonder whether additional cognitive mechanisms were involved in inhibition, as this is always the case when studying executive functions (EFs) such as working memory, inhibition, planning, cognitive flexibility. Part of the issue consists in the difficulty to define and differentiate between EFs, which leads to the well-known problem of ‘impurity’. Indeed, many tasks involve more than one EF and it is challenging to develop a paradigm where the participant would use just one (Miyake, 2012). Notably, inhibition itself is a multifaceted process with subcomponents. That is the reason why we always specify that we are investigating a specific inhibitory process, namely the ability to inhibit a prepotent motor response. Therefore, we administered an adapted version of the Go/No-Go task, built upon the configurations that allow us to evoke prepotent motor activity and tap significantly more on the inhibitory process of interest than on other EFs (Wessel, 2017). We are open to further discuss these challenging topics with the reviewers.

R2: In the present study, the authors examined how two different reaching tasks, a prepotent (dominant) task and an alternate (non-dominant) task, influenced motor planning and online adjustments. Motor planning and online adjustments were inferred from a number of kinematic measures; reaction time (RT), movement duration (MD), and time to peak velocity percentage (TPV%). Parts of these measures were extrapolated from 3-D accelerometer data. The authors found that, while the absolute measures of RT and MD were overall longer during non-dominant reaching, the relative measure of TPV% occurred later in the non-dominant task compared to the dominant. The authors suggest that motor planning was more valuable to participants than online adjustments; the prepotent task had to first be inhibited and an alternate task selected. This is an interesting paper, and I do believe that it adds valuable information to current literature. However, I have some major concerns regarding the current state of the manuscript that I feel should be addressed prior to publication. My major, and minor, concerns are listed below.

Major Concerns

1. My first major concern involves the formatting of the manuscript. In its current form, the manuscript is not presented in traditional manuscript format. The introduction is extremely long, due largely to the fact that it contains a great deal of information better suited to the methods section. The introduction is also lacking hypotheses. There is no mention of statistical analyses in the methods section. Rather, statistical methods are scattered throughout the results section. I normally do not comment on writing style or paper format as this is a largely subjective point that says nothing of the scientific quality of the work. However, in its present form, the manuscript is confusing and difficult to read, making it a challenge to locate pertinent information and decipher the findings of the study. The manuscript needs to be revised to improve readability. Specifically:

a. The introduction should concisely establish a framework related to the current field of work and state the problem/gap that currently exists in the literature. This should be followed by a clear statement of the study’s purpose and the hypotheses. The introduction could be easily cut down to half of its current size.

b. Statistical analyses/tests should be discussed at the end of the methods section, not in the results section. Currently, the majority of the results section are actually method details, not study findings.

A: Our gratitude to the reviewer for this thoughtful and accurate revision. We agree with his/her suggestions and therefore:

- Included a proper “aims and hypothesises” section

- Described the statistical approach in a proper section of the method part

We also tried to make the introduction sounder and more concise and moved the methodological details in the method section. We are open to make further changes if needed.

R: 2. A second reason why I find the study’s introduction so confusing is that it seems to be trying to accomplish two very different things simultaneously. On one hand, this appears to be a motor planning/motor control study, and on the other, it seems to be a validation study for wearable 3-axis accelerometers. For instance, in the final line of the introduction, the authors state: “In the following, the methods and results, as well as considerations concerning the feasibility, reliability and validity of such an approach are discussed.” However, I don’t see how this is a validation study as the TPV% values were never compared to a gold standard, such as values obtained from motion capture setups. Also, this seems to already have been accomplished by Cahill-Rowley and Rose in 2017 (cited by the study authors), who validated peak velocity magnitude and timing from accelerometers against Motion Analysis Corporation’s setup. Instead, the authors seem to have based the accuracy of their own data on how close it relates to the work of other studies. For example, in the discussion section on lines 344 – 352, the authors make numerous statements such as: “On the other hand, those related to the velocity shape over time seemed to be valid indices.” I am confused by this entire section. It should either be valid or not valid, and if it is valid, what was it validated against? To address this point, the authors should clearly state whether this was a validation study or not. If it is a validation study, how was TPV% validated, and what has this work added to the field that has not already been determined by previous research? (Example, Cahill-Rowley and Rose 2017).

A: We thank the reviewer for pointing out these limitations in our manuscript. We are aware that, although primarily focusing on motor planning/control aspects, we also questioned the validity of our accelerometer. We also agree that we are not validating our tool, as we are not comparing its data with those collected through gold standard motion capture systems. This was done by Cahill-Rowley and Rose 2017, whose study was already cited in our manuscript and is now further described in the apparatus section of methods:

“In particular, Cahill-Rowley and Rose [19] analysed human reaching kinematics through both inertial sensors and gold standard motion capture systems. The two methods provided consistent measures of displacement, peak velocity magnitude and timing. In light of this encouraging evidence, the time is ripe for the use of low-cost accelerometers to investigate distinct neuropsychological mechanisms beneath action selection.”

Despite the encouraging means in literature, we directly evaluated the reliability and validity of the calibrated and preprocessed acceleration and the computed velocity values. We did so from a mathematical perspective, that is described in detail in S3 Appendix (which is frequently referenced in the manuscript). 

In particular, we considered the data collected by an experimenter not belonging to the cohort involved in our trials. We measured the distance between the sensor and where the response keys appear on the touchscreen, corresponding to the actual hand displacement required to reach the screen. We then compared such displacement to the one computed from the acceleration data (estimated displacement). We calculated the estimated displacement in three different ways. With all these mathematical methods, the mean estimated displacement is distant from the actual displacement and the standard deviations are quite high. Concluding, we were not able to confirm nor disprove the reliability and validity of acceleration and velocity magnitudes obtained from our setting. Nevertheless, we highlight in the manuscript that these limitations do not seem to affect the peak velocity timing (“when” in time, e.g., time to peak velocity), that were fruitfully used to analyse the participants’ performance.

We believe that the Appendices would not only increase the transparency and replicability of our protocol, but also capture the interest of readers from data analysis and informatics fields. Overall, we agree that assessing the validity and reliability of the accelerometer was not our primarily goal and reduced the focus on that aspect within the manuscript. We are open to further discuss the Appendix content with the reviewers and make further clarifications. 

R: Minor Concerns

3. Although the manuscript is certainly readable in its current form, there are quite a few errors with grammar and sentence structure that should be corrected.

A: We thank the reviewer for pointing that out and have indeed proofed this manuscript thoroughly for typos and English errors before this resubmission.

R: 4. Lines 57 – 59: The authors highlight in this sentence that wearable IMUs with built in accelerometers offer promising potential for the future. The Cahill-Rowley and Rose, 2017 paper is referenced here, but no reference is made to what they actually did in their study. The overall introduction seems to be implying that the present study is the first attempt to validate the use of wearable accelerometers, but clearly, papers already exist on this topic. Again, it is not evident to me what the novelty of this study is in regards to the use of wearable accelerometers.

A: Our gratitude to the reviewer for remarking this issue, which has now been addressed by further describing Cahill-Rowley and Rose’s paper. Although wearable accelerometers have been previously adopted by researchers, we believe the novelty of our study is the application of this technology to the investigation of a specific neuropsychological mechanism (i.e. motor inhibition).

R: 5. Line 63: This is the first time I’ve seen a separate section all to itself included within an introduction, and as stated in my first major comment, I am confused as to why it has been included. Part of this section outlines the concept of investigating the nature of motor planning versus online adjustments using “Go” cue experiments. This could all be included in the main body of the introduction. The rest of the section is spent defining kinematic measures such as RT and MD, which are very standard kinematic measurements that shouldn’t warrant this much text to explain. As stated in my first major comment, the introduction could be easily reduced by 50%.

A: We thank the reviewer for his/her suggestions, and we have made changes accordingly. In particular, the introduction is just one (shorter) section in the current version. However, we believe that defining standard and vastly adopted kinematic measures is necessary as equivalent ones are frequently used with different labels, whereas identical labels are used to refer to different indices. For example, different researchers or research fields define Reaction Time either as we do, or as the entire response time (our RT+ Movement Duration). Moreover, the definition of these indices in the introductive section is not an end in itself, but rather preparatory for furthering that each measure is a proxy of distinct neuropsychological processes. Moreover, the other reviewer suggested to break things down even simpler for non-motor control readers that may not be familiar with specific terms. Therefore, we tried not to give too many kinematic definitions for granted. We are open to further discuss this aspect with the reviewers.

R: 6. Lines 116-118: As stated in my second major comment, please explain what is meant by validation. The study seems to be trying to validate the use of wearable 3-axis accelerometers, but I don’t see how this validation was accomplished.

A: We thank the reviewer for strengthening the concern about this important aspect. We replied to the reviewer’s second major comment with further details. We hope that the current version of the manuscript fulfils this point.

R: 7. Lines 121 – 122: An uneven number of females and males participated in this study. Is it possible that sex could affect the kinematic measures, particularly TPV%? Might males and females employ unique strategies when performing a non-dominant task? In other words, is it possible that females might rely more or less on motor planning time than males? The same goes for online adjustments. Was sex accounted for in the statistical tests?

A: We appreciate and agree with the reviewer’s concern about potential gender differences, which were not a main interest of the present work. We did not account for gender in the statistical tests, as the sample was not appropriate to do so. We now acknowledge the unbalanced sample while discussing about limitations and highlighted that future research could expand on the interesting topic of gender differences.

R: 8. Line 170: Please explain why trials longer than 2000 ms were omitted from the study. Given that movement duration was longer for the non-dominant reaching tasks, I assume that more trials were omitted in these tasks than the dominant (prepotent) tasks. If so, would this not diminish the difference between the two reaching tasks?

A: We thank the reviewer for asking about the rational for establishing this time limit. Indeed, we were interested in eliciting fast responses and decided the 2000 ms threshold a priori (which is quite common in the extant literature, although the trial duration is quite variable across different studies - Wessel, 2017). After 2000 ms from the start of the trial, the target stimulus disappeared, and the participant was no longer allowed to respond. Therefore, we marked those trials as “omission” cases. On the heels of your question, we specified the number of errors and omissions that participants made in each type of task, as well as the number of anticipations. In particular, omissions occurred 107 times (78 in the dominant condition that was presented the 75% of times). We can say that, proportionally, there is no meaningful difference among the number of omissions in the two conditions.

R: 9. Lines 223 – 225: I may be confused in this, but from what I can gather, the authors initially calculated velocity from acceleration using a trapezoidal integration function. There was a concern that such a function could lead to additional errors if a bias existed. Specifically, it could create a false peak in velocity. Thus, the authors applied a detrend function. As far as I can tell, velocity magnitude was not reported in this study, so why was this detrend function deemed necessary? TPV% was the reported measure, and I don’t believe integration would skew the time domain.

A: We thank the reviewer for giving us the chance to further clarify this aspect. As the reviewer says, we do not report velocity magnitudes in light of the considerations furthered in the Appendices. However, we analysed the TPV% that is the velocity timing (the percentage of time spent from the movement start to maximum peak velocity). Although the time domain was not skewed by the integration function, the latter could create a false peak, as visible in Figure 2 (x component and magnitude of velocity) and in S2 Appendix (figure 2a). In that case (of additional numerical errors), the velocity curve continues to increase instead of presenting the expected bell shape. Therefore, from a mathematical point of view, the (maximum) peak for the TPV calculation would be at the end rather than at around 40% of movement time. In other words, although the “true” peak is still there, another bigger peak would be detected as the maximum one. As explained in S2 Appendix, the detrend function overcame this issue. To the end of making these points clear, we have now rephrased that part of the manuscript and the figure caption.

R: 10. Lines 250 – 253: TPV% measures were excluded if they occurred outside of 5 and 95% of the trial’s movement time. The provided rationale is that anything outside of 5 or 95% represents technical failure rather than true human movement. A total of 59 trials were omitted from the present study, indicating that technical failure did in fact occur in some trials. I find this point concerning. How do the authors know that trials within the 5-95% represented true human movement and weren’t also influenced by technical errors/failures? For instance, if I picture myself performing this reaching task myself, I would find values of 6 and 94% equally unlikely to be the true time of peak velocity as values outside of 5-95%.

A: We thank the reviewer for touching on this crucial point. We agree that the 5-95% range is conventional and somehow arbitrary as it always happens with statistical thresholds. For example, if you think about p-values, “God loves the .06 nearly as much as the .05” (Rosnow & Rosenthal, 1992). We believe that descriptive statistics of TPV% (mean and standard deviation) will provide a good sense of the variability of our data. For the sake of clarity and transparency, we have also included the minimum and maximum of all our collected measures.

R: 11. Again, I could be missing this simply because of some odd formatting choices, but were figure captions included anywhere in the initial submission? Typically, these are submitted either directly with the figures themselves, or they are included as a list either before of after the reference list. I didn’t see these included anywhere, which makes interpretation of the graphs difficult. For instance, on lines 223-225, the authors state that the bias is visible in Figure 2, but since no caption is included with the figure, it is not clear what the authors are referring to. The figures do not currently speak for themselves, and I do not see a clear sign of any bias in Figure 2.

A: We are sorry to appreciate that the reviewer had a hard time checking our figures. We believe this is due to the journal guidelines for submission, that required us to fit the captions within the manuscript and separately upload the figures alone. We agree that alternative formatting choices would have been smarter. The reviewer should be able to find all captions within the manuscript. For instance, the caption for Figure 2 states:

“Velocity signals of a trial where the error due to the acceleration bias is visible in both the x component and the magnitude (increasing monotonous curves that do not represent the expected bell shape of velocity)”

Moreover, we clarified the meaning of bias in the manuscript:

“Notably, the application of an integration function could lead to an incremental numerical error due to a possible bias (i.e., additive noise) present in the acceleration, visible in Figure2, whereby the x component and magnitude of velocity present increasing monotonous curves rather than the expected bell shape”

Additional notes:

We have addressed the journal requirements and

1. Ensured that our manuscript meets PLOS ONE's style requirements, including those for file naming and references formatting. 

2. Provided further details on participant recruitment in the Methods section.

3. Changed "female” or "male" to "woman” or "man" as appropriate, when used as a noun.

We confirm our Data Availability statement, and we will provide repository information for our data at acceptance.

---

## [Decision Letter · Decision Letter 1]

10 Jun 2021

PONE-D-20-39385R1

Reaching to inhibit a prepotent response: A wearable 3-axis accelerometer kinematic analysis

PLOS ONE

Dear Dr. Farroni,

Thank you for submitting your manuscript to PLOS ONE. After careful consideration, we feel that it has merit but does not fully meet PLOS ONE’s publication criteria as it currently stands. Therefore, we invite you to submit a revised version of the manuscript that addresses the points raised during the review process.

Please address reviewer 2's remaining concern by making the needed changes to the manuscript.

We look forward to receiving your revised manuscript.

Kind regards,

Bernadette Ann Murphy, PhD

Academic Editor

PLOS ONE

Journal Requirements:

Additional Editor Comments (if provided):

Please address reviewer 2's remaining concern by making the needed changes to the manuscript.

Reviewers' comments:

Reviewer's Responses to Questions

**Comments to the Author**

1. If the authors have adequately addressed your comments raised in a previous round of review and you feel that this manuscript is now acceptable for publication, you may indicate that here to bypass the “Comments to the Author” section, enter your conflict of interest statement in the “Confidential to Editor” section, and submit your "Accept" recommendation.

Reviewer #1: All comments have been addressed

Reviewer #2: (No Response)

2. Is the manuscript technically sound, and do the data support the conclusions?

Reviewer #1: Yes

Reviewer #2: Yes

3. Has the statistical analysis been performed appropriately and rigorously? 

Reviewer #1: Yes

Reviewer #2: Yes

4. Have the authors made all data underlying the findings in their manuscript fully available?

Reviewer #1: Yes

Reviewer #2: Yes

5. Is the manuscript presented in an intelligible fashion and written in standard English?

Reviewer #1: Yes

Reviewer #2: Yes

6. Review Comments to the Author

Reviewer #1: The authors have addressed all of my previous concerns. However, the correction to the abstract (noted within the response letter) did not appear in the revised version of the manuscript so I'd recommend that the authors double check that correction prior to publication.

Reviewer #2: Overall, the manuscript has been significantly improved, and most of my comments have been addressed. The authors have taken my initial feedback and made a number of important changes to their paper. I only have one remaining concern.

In my initial comments, I raised a concern over how the TPV% data was processed. The paper states that TPV% measures were excluded if they occurred outside of 5 and 95% of the trial’s movement time; values less than 5% or greater than 95% would represent technical failure rather than true human movement. I asked how the authors could be confident that values within 5 and 95% were valid measurements, and they responded with a statistics analogy to the arbitrary alpha level of .05. Yes, .05 is indeed arbitrary, but my concern has nothing to do with statistics. It has to do with how the data was processed, and data processing should not be arbitrary. I will try and clarify my earlier comment here.

My concern centers around the authors’ use of “technical failure.” To me, technical failure would mean that something went wrong with either the equipment, the software, data processing, or a combination of all three, which ultimately led to an incorrect, or erroneous, TPV% value. The current solution was to simply omit values less than 5% or greater than 95%, but as the authors themselves state, these are arbitrary numbers. My concern is, how do you know that values within 5-95% weren’t also erroneous? For example: in a single trial, a person might have reached their TPV% at 50% of the total movement time, but the value that was actually calculated came out to be 87% (due to a technical failure). This value of 87% is an error, but since it falls within the 5-95% boundaries, it will be included in the data set. How do the authors know that this wasn’t happening?

If I were to draw a simple analogy, electromyography (EMG) collections typically use a band pass filter (example: 10-500 Hz; anything below 10 Hz and anything above 500 Hz will be omitted from the signal). This serves to remove any low frequency noise, such as movement, and high frequency noise from the signal. However, electrical noise with a frequency range within 10-500 Hz can still contaminate the signal. How do the authors know that these “technical failures” (i.e. noise) weren’t present in the 5-95% acceptable range?

7. PLOS authors have the option to publish the peer review history of their article (what does this mean?). If published, this will include your full peer review and any attached files.

Reviewer #1: No

Reviewer #2: No

---

## [Author Response · Author response to Decision Letter 1]

25 Jun 2021

PONE-D-20-39385R1

Reaching to inhibit a prepotent response: A wearable 3-axis accelerometer kinematic analysis

PLOS ONE

25 th June 2021

Response to Reviewers

A: (Answer)

Journal Requirements:

A: We would like to thank the journal for this note. We then reviewed all the reference list. In particular, we found in S1 Appendix bibliography a citation to a paper found at the moment of submission which is now no longer available (old citation number 6 in S1 Appendix: “Thorncroft, G. How precise is earth’s gravity? https://web.calpoly.edu/~gthorncr/ME302/documents/AccuracyofGravity.pdf.”). This was a paper that referred to the approximation of gravitational constant g as g=9.80665m/s^2. Considering this fact and analysing the literature we then decide to remove this reference, since g=9.80665m/s^2 is a standard approximation for gravitational constant and a reference is not necessary. In addition, we found some missing information, such as page number, in a citation of the main bibliography (old citation number 7: “Jeannerod, M. and Biguer, B. (1982). Visuomotor mechanisms in reaching within extrapersonal space.”). As it is a chapter of a book which seems to be no longer distributed, and the reference does not have a central role for our work, we decided to remove it in order not to unnecessarily burdening the introduction.

Moreover, we also proofread the entire manuscript and corrected or reorganised a couple of sentences in the Abstract and Discussion sections in order to make the writing more fluid without obviously changing its content and meaning.

Reviewer #1: The authors have addressed all of my previous concerns. However, the correction to the abstract (noted within the response letter) did not appear in the revised version of the manuscript so I'd recommend that the authors double check that correction prior to publication.

A: Our apologise for this mistake. We have not corrected the abstract as previously stated.

Reviewer #2: Overall, the manuscript has been significantly improved, and most of my comments have been addressed. The authors have taken my initial feedback and made a number of important changes to their paper. I only have one remaining concern.

In my initial comments, I raised a concern over how the TPV% data was processed. The paper states that TPV% measures were excluded if they occurred outside of 5 and 95% of the trial’s movement time; values less than 5% or greater than 95% would represent technical failure rather than true human movement. I asked how the authors could be confident that values within 5 and 95% were valid measurements, and they responded with a statistics analogy to the arbitrary alpha level of .05. Yes, .05 is indeed arbitrary, but my concern has nothing to do with statistics. It has to do with how the data was processed, and data processing should not be arbitrary. I will try and clarify my earlier comment here.

My concern centers around the authors’ use of “technical failure.” To me, technical failure would mean that something went wrong with either the equipment, the software, data processing, or a combination of all three, which ultimately led to an incorrect, or erroneous, TPV% value. The current solution was to simply omit values less than 5% or greater than 95%, but as the authors themselves state, these are arbitrary numbers. My concern is, how do you know that values within 5-95% weren’t also erroneous? For example: in a single trial, a person might have reached their TPV% at 50% of the total movement time, but the value that was actually calculated came out to be 87% (due to a technical failure). This value of 87% is an error, but since it falls within the 5-95% boundaries, it will be included in the data set. How do the authors know that this wasn’t happening?

If I were to draw a simple analogy, electromyography (EMG) collections typically use a band pass filter (example: 10-500 Hz; anything below 10 Hz and anything above 500 Hz will be omitted from the signal). This serves to remove any low frequency noise, such as movement, and high frequency noise from the signal. However, electrical noise with a frequency range within 10-500 Hz can still contaminate the signal. How do the authors know that these “technical failures” (i.e. noise) weren’t present in the 5-95% acceptable range?

A: We would like to thank the reviewer for his/her careful considerations, and for clarifying this doubt about our choice to exclude TPVs outside the 5-95% range. We apologize for not fully understanding this question, which we will now attempt to answer more adequately. There seems perhaps to be some misunderstanding about the difference between data preprocessing and analysis.

Taking up the analogy with the EMG data, we can say that the bandpass filters that remove noise from the signal correspond to what we did in the preprocessing phase of the raw acceleration values collected by the accelerometer, as reported in detail in S1 Appendix. This preprocessing phase was meant to remove potential noise.

From the resulting preprocessed data, we computed the TPV index, which represents participants’ performance in each trial. Our apologizes for any misunderstanding caused by our improper use of “technical failures”, which primarily referred to “numerical errors, in cases where the detrend function was not sufficient to remove their effect on the signal”, whereby “numerical errors” refer to those resulted from the application of an integration function due to a possible bias (i.e., additive noise) present in the acceleration (as written in Section “Materials and methods – Kinematics measures”). Moreover, as in all cases of human performance measurement, the TPV is variable and potentially resulting from human errors (e.g., in the cases where participants made extra-task movements and did not precisely keep following the instructions). More generally, we decided to remove the whole observations (not part of the acceleration signals) with a TVP outside the 5-95% range. Indeed, based on the existing literature, we can expect TPVs around the average value of 50% of movement duration. Therefore, those smaller than 5% and greater than 95% seem unlikely related to the task-related human reaching movements that are required by our specific task and paradigm (an adapted Go/No-Go paradigm). 

In order to clarify this aspect, we rephrased this part of the manuscript as it follows:

Ultimately, we aimed to exclude possible extreme TPV% values that would be due to numerical errors, in cases where the detrend function was not sufficient to remove their effect on the signal. Moreover, we aimed to remove those observations with TPV% values that were unlikely related to task-related human reaching movements, but rather potentially ascribable to extra-task movements. For these reasons, the a-priori inclusion criteria for valid TPV% values comprehended those between 5% and 95%. At the end of this procedure, we excluded 59 out of 2,962 trials. 

As far as we correctly understand your comment, we are not able to check whether additional extra-task movements occurred and influenced the accepted TPVs, which is true for many psychology experiments, whereby average data are meant to flatten the effect of potential human-related noise. However, your suggestions made us wonder whether, in future studies, we might address this issue by including video recordings and offline coding of the experimental sessions, to further check for visible cases of extra-task movements. We are used to do so when testing infants and children, whose performance is particularly variable. We did not employ video recordings with the present study, as neurotypical adults are usually quite good at following experimental instructions and procedures. We now acknowledge this interesting point in the Discussion:

Moreover, future studies might include video recordings and offline coding of the experimental sessions, thus checking for potential cases where participants show extra-task movements that could result in anomalous trials.

Notably, the exclusion of extreme TPVs made us remove only 59 out of 2.962 trials (less than the 2% of observations). To further explore the effect of removing TPVs out of the 5-95 range, please see below the variable distributions (a) before and (b) after (equal to the one reported in Figure 3 of the manuscript) removing the TPVs outside the 5-95% range. Notably, the removal of extreme TPVs did not affect the distribution shape nor the difference between values observed in the conditions of interest (Dominant vs Non-dominant). Indeed, when we run the model comparison analysis (c) before and (d) after removing the TPVs outside the 5-95% range, the Condition effect appears robust. In conclusion, we are confident that the present manuscript presents robust and valuable results, and the actual limitations are properly described to inform future steps.

*Figures are fully visible in the file "Response to Reviewers" that we send along with this re-submission.

---

## [Editor Report · Decision Letter 2]

29 Jun 2021

Reaching to inhibit a prepotent response: A wearable 3-axis accelerometer kinematic analysis

PONE-D-20-39385R2

Dear Dr. Farroni,

We’re pleased to inform you that your manuscript has been judged scientifically suitable for publication and will be formally accepted for publication once it meets all outstanding technical requirements.

Kind regards,

Bernadette Ann Murphy, PhD

Academic Editor

PLOS ONE
---

## [Editor Report · Acceptance letter]

8 Jul 2021

PONE-D-20-39385R2 

Reaching to inhibit a prepotent response: a wearable 3-axis accelerometer kinematic analysis 

Dear Dr. Farroni:

I'm pleased to inform you that your manuscript has been deemed suitable for publication in PLOS ONE. Congratulations! Your manuscript is now with our production department. 

Kind regards, 

on behalf of

Dr. Bernadette Ann Murphy 

Academic Editor

PLOS ONE